

# Individual and combined effect of organic eutrophication (DOC) and ocean warming on the ecophysiology of the Octocoral *Pinnigorgia flava*

Edoardo Zelli[1,2,*], Susana Marcela Simancas-Giraldo[1,*], Nan Xiang[1,3], Claudia Dessì[1,4], Nadim Daniel Katzer[1], Arjen Tilstra[1] and Christian Wild[1]

[1] Marine Ecology Department, Faculty of Biology & Chemistry (FB 2), University of Bremen, Bremen, Germany
[2] School of Science, University of Waikato, Tauranga, New Zealand
[3] Helmholtz Centre for Polar and Marine Research, Alfred Wegener Institute, Bremerhaven, Germany
[4] Dipartimento di Scienze della Vita e dell'Ambiente, University of Cagliari, Cagliari, Italy
* These authors contributed equally to this work.

Corresponding author
Edoardo Zelli,
zelli.edoardo@gmail.com

## ABSTRACT

Dissolved organic carbon (DOC) enrichment and ocean warming both negatively affect hard corals, but studies on their combined effects on other reef organisms are scarce. Octocorals are likely to become key players in future reef communities, but they are still highly under-investigated with regard to their responses to global and local environmental changes. Thus, we evaluated the individual and combined effects of DOC enrichment (10, 20 and 40 mg $L^{-1}$ DOC, added as glucose) and warming (stepwise from 26 to 32 °C) on the widespread Indo-Pacific gorgonian *Pinnigorgia flava* in a 45-day laboratory experiment. Oxygen fluxes (net photosynthesis and respiration), as well as Symbiodiniaceae cell density and coral growth were assessed. Our results highlight a differential ecophysiological response to DOC enrichment and warming as well as their combination. Individual DOC addition did not significantly affect oxygen fluxes nor Symbiodiniaceae cell density and growth, while warming significantly decreased photosynthesis rates and Symbiodiniaceae cell density. When DOC enrichment and warming were combined, no effect on *P. flava* oxygen fluxes was observed while growth responded to certain DOC conditions depending on the temperature. Our findings indicate that *P. flava* is insensitive to the individual effect of DOC enrichment, but not to warming and the two stressors combined. This suggests that, if temperature remains below certain thresholds, this gorgonian species may gain a competitive advantage over coral species that are reportedly more affected by DOC eutrophication. However, under the expected increasing temperature scenarios, it is also likely that this octocoral species will be negatively affected, with potential consequences on community structure. This study contributes to our understanding of the conditions that drive phase shift dynamics in coastal coral reef ecosystemds.

## INTRODUCTION

Gorgonians (Octocorallia; Alcyonacea [= Gorgonians]) have, in general, a flexible internal gorgonin skeleton which in some species take on tree- or bush-like forms that can reach considerable sizes (up to >1 m in height). This non monophyletic group differs greatly from the most studied Scleractinian corals (Hexacorallia), also known as hard or reef-building corals, because the latter have a hard, calcium-based skeleton which, in most cases, constitutes the very foundation of the reef ecosystem (*Roberts & Hirshfield, 2004*). Nevertheless, both scleractinian and gorgonians are generally considered ecosystem engineers due to the key ecological role they carry out in ecosystem functioning (*Jones, Lawton & Shachak, 1994*; *Rossi et al., 2017*). In particular, gorgonians typically have an arborescent shape that forms three-dimensional structures which increase environmental complexity. While gorgonians are not a recognized taxonomically valid group, this term is useful for referring to soft corals that have a skeletal axis composed of gorgonin (*McFadden, Sánchez & France, 2010*), and a more extensive three-dimensional structure as a result of their axial skeleton. These organisms often create underwater forests that foster biodiversity either as a substrate for epifaunal communities or by acting as nursery areas for a large number of species (*Sánchez, 2016*; *Rossi et al., 2017*). In tropical coral reef ecosystems, the ecological functioning of many gorgonian species is strongly dependent on the relationship between the coral host, its associated microbes and associated Symbiodiniaceae algae (*i.e.*, collectively the coral holobiont), which provide the foundation for their ecological success, as they do for many other tropical coral species. In this mutualistic association, the coral host provides inorganic nutrients in exchange for photosynthetically fixed carbon (photosynthates) and amino acids translocated from the Symbiodiniaceae (*Muscatine & Porter, 1977*), which fuels gorgonian growth (*Ramsby et al., 2014*).

Anthropogenic impacts on a local and a global scale can threaten the coral-Symbiodiniaceae symbiosis (*Pogoreutz et al., 2017*; *Rädecker et al., 2021*; *Tilstra et al., 2017*). Increased water temperatures can induce malfunctioning of the Symbiodiniaceae photosynthetic apparatus leading to a reduction in Symbiodiniaceae cell density and subsequently photosynthetic activity (*Weis, 2008*; *Fitt et al., 2001*; *Hughes et al., 2017*; *Hughes, Kerry & Simpson, 2018*). Moreover, extended periods of high temperatures generally lead to coral mortality due to physiological damage and impaired metabolism (*Glynn, 1984*; *Berkelmans et al., 2004*). These negative impacts of increased temperatures are described in both hard corals (*Santos et al., 2014*; *Cardini et al., 2016*; *Hughes, Kerry & Simpson, 2018*; *Ziegler et al., 2019*) and octocorals, such as gorgonians (*Lasker, 2003*; *Harvell et al., 2001*; *Prada, Weil & Yoshioka, 2010*). Gorgonians often respond similarly compared to the more extensively studied hard corals—*i.e.*, display bleaching processes whereby large extents of the coral rapidly pale through loss of their algal endosymbionts

*via* destabilization of the coral–algal symbiosis. Coral bleaching may lead to diminished photosynthetic activity and eventually to mortality when prolonged over long periods (*Brown, 1997*; *Suggett & Smith, 2020* and references therein).

Yet, coral resilience towards temperature-induced impacts can be simultaneously affected by other stressors (*Hughes et al., 2003*; *Brodie et al., 2011*), including complex combinations of stressors arising from global climate change and local degraded water quality. The presence of humans in the proximity of coral reefs can result in an elevated input of nutrients into reef waters. Nutrients associated with human activities, *i.e.*, particulate and dissolved inorganic and organic matter, can enter reef ecosystems *via* riverine influx, diffuse discharge, or as aeolian dust (*Cuet, Naim & Faure, 1988*; *Fabricius & De'ath, 2004*; *Wooldridge, 2009*; *Wagner, Kramer & Van Woesik, 2010*; *Brodie et al., 2009*, *2012*; *Weber et al., 2012*). Some corals may benefit from particulate organic matter enrichment because it enhances feeding rates and growth, providing even higher competitive advantages, especially for species more dependent on heterotrophic filter feeding. A consequence of this is a potential community shift from corals that can grow at extremely low food concentrations to more heterotrophic and less diverse coral communities (*Fabricius, 2005*). However, anthropogenic eutrophication of coastal waters has also been linked to a decline in coral cover (*Bednarz et al., 2012*; *Wiedenmann et al., 2013*; *Pogoreutz et al., 2017*).

Land sourced runoff containing elevated nutrient concentrations may result in a wide range of impacts on hard coral communities (*Grigg, 1995*; *Ward & Harrison, 2000*; *Koop et al., 2001*; *Loya et al., 2004*; *Fabricius & De'ath, 2004*; *Fabricius & De'ath, 2004*; *Fabricius, 2007*); including reduced recruitment (*Loya et al., 2004*; *Fabricius, 2005*), modified trophic structures (*Fabricius & De'ath, 2004*), altered biodiversity (*Woesik, Tomascik & Blake, 1999*), and increased mortality (*Ward & Harrison, 1997*; *Harrison & Ward, 2001*; *Kline et al., 2006*). Under extreme situations, such impacts can result in the collapse of the coral community (*Smith et al., 2006*). Furthermore, experiments on hard corals indicate that increased nutrient levels can reduce tolerance to heat stress (*Wooldridge, 2009*; *Wagner, Kramer & Van Woesik, 2010*; *Cardini et al., 2015*), which assigns critical importance to local management of water quality to mitigate the pressure induced by global climate change (*Webb et al., 1975*; *Hoogenboom et al., 2012*). Thus far, studies assessing the impacts of nutrient enrichment on coral reefs have primarily focused on hard corals (*e.g.*, *Wiedenmann et al., 2013*; *Vega Thurber et al., 2014*; *Cardini et al., 2015*) with very few studies on octocorals. Gorgonians may have a capacity to cope with inorganic nutrient enrichment (*Fleury et al., 2004*; *McCauley & Goulet, 2019*), while organic matter fluxes and metabolic activity in other octocorals may be negatively affected (*Bednarz et al., 2012*; *Baum et al., 2016*; *McCauley & Goulet, 2019*).

Nevertheless, water quality has many components which have not been as widely studied and deserve more attention, such as dissolved organic carbon (DOC). In hard corals, DOC enrichment can cause a breakdown of the coral-Symbiodiniaceae symbiosis (*Pogoreutz et al., 2017*), similar to thermally stressed corals (*Rädecker et al., 2021*). This process is intimately linked with nitrogen (N) availability (*Wiedenmann et al., 2013*; *Wooldridge, 2013*; *Vega Thurber et al., 2014*; *Rädecker et al., 2021*). DOC enrichment may

stimulate the fixation of atmospheric $N_2$ by increased microbial—*i.e.*, diazotrophic-activity, increasing N concentration and triggering the rapid uptake of N by Symbiodiniaceae (*Pogoreutz et al., 2017*). This may eventually lead to phosphorus (P) starvation and a stoichiometric shift in the N:P ratio, causing the photosynthetic apparatus to malfunction resulting in the onset of bleaching (*Tchernov et al., 2004*; *Wiedenmann et al., 2013*; *Wooldridge, 2013*). Other studies showed hard coral mortality under DOC enrichment treatments and similar experimental duration (*Kline et al., 2006*; *Kuntz et al., 2005*). Recent studies assessing the individual and combined effects of DOC enrichment and increased water temperatures on the fleshy, pulsating octocoral *Xenia umbellata* reported that DOC enrichment had a positive effect on its heat tolerance when functional and ecological variables, *i.e.*, pulsation, were considered (*Vollstedt et al., 2020*). DOC enrichment also did not have a significant effect on the ecophysiology (oxygen production, consumption, and growth) of *X. umbellata* suggesting that certain octocorals may be more resistant to individual DOC enrichment than hard corals (*Simancas-Giraldo et al., 2021*). In line, *X. umbellata* showed decreased gross photosynthetic activity at 28 and 30 °C but still positive net photosynthesis at 32 °C displaying certain degree of resistance to elevated temperatures.

Despite empirical evidence showing that octocorals are becoming more abundant, displaying more significant functional roles than in the past (*Lenz et al., 2015*; *Ruzicka et al., 2013*), and potentially representing a "new normal" for some coral reefs, they remain largely under-investigated (*Lasker et al., 2020*). In particular, how the combined effects of local (*e.g.*, organic eutrophication) and global factors (*e.g.*, warming) influence the ecophysiological responses in gorgonians is still largely overlooked. Thus, in this study, we investigated how the individual and combined effects of DOC enrichment and increased temperatures affected the ecophysiology of the Symbiodiniaceae-associated gorgonian *Pinnigorgia flava* (*Nutting, 1910*). Following previous studies on nutrient addition and thermal stress (*Fabricius et al., 2013*; *McCauley & Goulet, 2019*), we assessed the effects of individual DOC concentrations and a subsequent stepwise increase in temperature on fragments of the gorgonian coral *P. flava* in a 45-day manipulative experiment. We hypothesized that the individual and combined effects of DOC enrichment and high temperatures would negatively impact this gorgonian's ecophysiology. Specifically, we expected coral photosynthesis, respiration activity and growth rates to significantly decrease under both individual and combined effects of DOC enrichment and increased temperature (with stronger negative effects under multiple stressors). As such, the present study assessed the effects of (1) organic eutrophication, *i.e.*, DOC enrichment, (2) increased water temperatures, and (3) a combination of both factors on the ecophysiology of the gorgonian *P. flava* by measuring photosynthesis and respiration activity *via* oxygen fluxes, Symbiodiniaceae cell densities and coral growth (*i.e.*, changes in surface area), over 45 days.

## MATERIALS AND METHODS

The experiment was carried out at the Marine Ecology laboratory of the Centre for Environmental Research and Sustainable Technology (UFT), University of Bremen, Germany.

## Experimental tank setup

Experimental design, methodologies, and the seawater parameters are described by *Vollstedt et al. (2020)* and *Simancas-Giraldo et al. (2021)*. In summary, our experiment was divided into two temporal stages consisting of 21 and 24 days, respectively. In the first experimental stage, 12 tanks (three tanks per treatment including three controls) were used to test the individual effects of three different DOC concentrations (low: 10 mg $L^{-1}$, medium: 20 mg $L^{-1}$ and high: 40 mg $L^{-1}$), while the control tanks were kept without DOC additions at environmental conditions (2 to 3 mg $L^{-1}$). During the second experimental stage, warming scenarios were implemented and four additional tanks were added as controls for the increased temperature treatments. In this stage, we tested the individual and simultaneous effects of warming by raising the water temperature following a stepwise increase from 26 to 32 °C (2 °C per week) in every tank except for the four tanks assigned as temperature controls. In further detail, 16 tanks were prepared in total for the experiments comprising this second experimental stage and ensured to have starting comparable conditions. Then, 12 of these tanks were used to accommodate the individual DOC treatments and the corresponding controls (*i.e.*, DOC controls) during the first stage, while the additional four tanks remained in wait for the start of the second stage of experiments. These additional tanks were installed and held with identical initial conditions to the ones in the DOC control tanks of the first stage and were employed as temperature controls during the second stage of the experiments, when all DOC treatments, including the DOC controls, were exposed to warming conditions.

Each experimental tank had a total volume of 60 L and consisted of two parts: a back part acting as a technical tank separated by a glass division from a front part which contained the experimental fragments. Both parts were connected by a pump and an outflow situated in the glass division between the frontal and back part that allowed water exchange between these two sections. Thus, each 60 L tank represented an independent closed system with its own circulation. Above each tank, a LED light simulated daylight conditions. All tanks were filled with artificial seawater (Tropic Marin® ZooMix Sea Salt) and kept at the same conditions as the tank from which parent colonies originated. One month prior to the experiments, water was cycled altogether with the parental colonies tank (*i.e.*, the maintenance tank) for a minimum period of 2 weeks, before closing connections between tanks making each of them independent. Seawater salinity was kept at 35 ± 0.6 ppt, pH of 8.2 ± 0.1, and temperature at 26.0 ± 0.3 C (mean ± SE) and exposed to a 12:12 light:dark period at constant light intensity (120.8 ± 10.2 µmol $m^{-2}$ $s^{-1}$), while additional chemical water parameters, such as the pH, KH, Ammonium ($NH_4^+$), Nitrite ($NO_2^-$), Nitrate ($NO_3^-$) and Phosphate contents ($PO_4^{3-}$) were measured and adjusted manually twice per week. A general summary on the chemical parameters through the experiment can be found in *Vollstedt et al. (2020)* while other relevant parameters such as salinity and pH can be found summarized by treatment in *Xiang et al. (2021)*. We additionally present details on the recorded mean values measured per tank through the experiment in our Material S1.
A parallel study on the soft coral *X. umbellata* was performed in conjunction with the current experiment. In particular, each experimental tank contained four frames, two holding $n = 10$ *P. flava* and two holding $n = 10$ *X. umbellata* fragments. The additional frames with *X. umbellata* fragments were used for different studies besides this one (*i.e.*, *Vollstedt et al., 2020*; *Simancas-Giraldo et al., 2021* and *Xiang et al., 2021*). Our selection of DOC treatment concentrations for this experiment was based on previous studies that manipulated glucose loading (*Kline et al., 2006*; *Pogoreutz et al., 2017*) alongside previous findings by *e.g.*, *Baum et al. (2016)*, where some octocoral species were shown to be able to inhabit highly eutrophicated zones in particular reefs. Untreated DOC tanks (2 to 3 mg $L^{-1}$) were employed as DOC control condition. The temperature treatments (28 °C, 30 °C, 32 °C) were selected to simulate the predicted rising ocean temperatures based on the 2018 IPCC report (*De Coninck et al., 2018*). DOC concentrations were measured using a Total Organic Carbon (TOC) analyzer (TOC-L CPH/CPN PC-Controlled Model; Shimadzu, Kyoto, Japan) twice a day, and adjusted by adding a standard solution containing D-Glucose anhydrous (purity: 99%; Thermo Fisher Scientific U.K. Limited, Loughborough, UK). The water temperature was measured daily and kept under stable conditions using a heater for each tank and salinity was kept steady by adding demineralized water to the system to compensate for evaporation.

## Experimental implementation

*P. flava* was molecularly identified by *Xiang et al. (2021)* and selected for this study based on its widespread occurrence in the Indo-Pacific (*Vargas et al., 2020*) and its relatively simple breeding and maintenance in experimental tanks (*Conci, Wörheide & Vargas, 2019*; *Vargas et al., 2020*; *Vargas et al., 2022*). The identity of the *P. flava* associated Symbiodiniaceae of our fragments was not confirmed through molecular means. However, since *P. flava* has been consistently reported to be associated with *Cladocopium sp.* (*Goulet, LaJeunesse & Fabricius, 2008a*), we inferred our *P. flava* fragments to be associated with a Symbiodiniaceae species within this genus. (see *Goulet, Simmons & Goulet, 2008b*; *LaJeunesse et al., 2018* and the "Symbiodiniaceae Style Guide" edited by Parkinson's Lab— https://www.thelifeaquatic.net/?page_id=292 for further details). We therefore did not expect the specific species of Symbiodiniaceae to play a significant role in the interpretation of our results.

During this study, approximately 280 fragments (2 ± 0.5 cm in length) of the gorgonian *P. flava* were propagated randomly from three clonal mother colonies (similar sized). The mother colonies initially originated from the Caribbean and were kept in a 420 L aquarium in the facility for more than 1 year prior to the start of our experiment. Except for water flow rates, we kept the same conditions of this aquarium for all the control tanks used for this experiment (Material S1). Each fragment was subsequently attached to calcium carbonate plugs Aqua Perfect frag plug for light grid/Round 1 cm (AP-7004-0) using coral glue (D-D AquaScape Construction Epoxy). Two plastic grids were used to fit a total of ten gorgonian fragments per grid. The fragments were then randomly assigned to their corresponding experimental tanks and allowed to acclimatize before the start of the experiment. A total of 240 *P. flava* fragments (20 fragments per tank × 12 tanks) were

employed during the first stage of the experiment (individual DOC addition). These fragments were acclimated in the experimental system for 5 days before the start of the first stage. The remaining 40 fragments were kept at ambient conditions in the maintenance tank for the first stage of the experiment. These fragments were then distributed evenly in the temperature control tanks and given 5 days to acclimate before the second stage of the experiment started.

## Quantification of oxygen fluxes

The oxygen ($O_2$) fluxes were calculated according to *Bednarz et al. (2012)*. Three fragments from each treatment were transferred to individual incubation glass chambers for measurements for oxygen quantification. In addition, one glass chamber per tank was filled solely with seawater to serve as a control to account for planktonic background metabolism (*i.e.*, control glass chamber). The starting $O_2$ concentration in each chamber was measured using a salinity-corrected $O_2$ optode sensor (FDO®925 Optical Dissolved Oxygen Sensor, range: 0.00 to 20.00 mg $O_2$ $L^{-1}$, accuracy: ±0.5% of the value, MultiLine® IDS 3430, WTW). All chambers were sealed airtight (without any air bubbles inside) and incubated twice per day, once for measuring $O_2$ production, *i.e.*, net photosynthesis, and once for $O_2$ consumption, *i.e.*, respiration. $O_2$ production was measured through incubations performed under light conditions, putting back the sampled glass chambers into the experimental tanks to keep the water temperature steady. Each glass chamber was opened, and end $O_2$ concentrations were measured as soon as the first incubation was concluded. The glass chambers were then closed immediately and subsequently incubated under complete darkness to measure $O_2$ consumption. During this measurement, the glass chamber was placed in darkened water baths which were completely clad with a handmade black coating and located inside a dark room. The temperature in the water baths was kept constant *via* thermostats, mirroring the temperatures in the corresponding experimental tanks. Thus, both water temperature and dark conditions were ensured during the dark incubations. The incubations lasted for approximately 2 h each. $O_2$ fluxes were subsequently calculated from these dark and light incubations, where $O_2$ initial concentrations were subtracted from the final concentrations and the results were normalized to the incubation time. The $O_2$ fluxes measured were further corrected by the background seawater control signal, subtracting the $O_2$ flux measured in the control glass chamber from the $O_2$ flux in the glass chamber containing the coral fragment. These corrections were further standardized by the incubation water volume and the calculated $O_2$ fluxes were finally normalized to the corresponding coral fragment surface area.

## Symbiodiniaceae density

Symbiodiniaceae cell counting was performed at the end of the experiment on day 45 by randomly selecting *P. flava* fragments ($n = 3$) from each aquarium treatment and cutting a ~1.5 cm tip. The branch tips were weighed in a four-digit analytical balance, to measure wet weight. Host tissue was then separated from its central gorgonin axis by mechanical removal of the tissue using a scalpel. Subsequently, simple mechanical movements were employed to separate the tissue from the axis. The resulting tissue slurry was collected in
2 mL Eppendorf tubes and homogenized mechanically with 1 mL of demineralized water using Eppendorf micropestles. Symbiodiniaceae cell density was subsequently quantified microscopically immediately after extraction using an improved Neubauer hemocytometer. Final counts were normalized to the corresponding coral fragment wet weight (*Forcioli et al., 2011*; *McCowan et al., 2011*; *Cardini et al., 2015*).

## Growth measurements

Growth of *P. flava* fragments was measured by image analysis using photographs taken continuously throughout the experiment ($n = 3$). The same fragments that were chosen for $O_2$ fluxes were photographed over time. Photographs of the fragments were taken once per week, always from a lateral view. The small fragments were glued as single unbranched fragments at the start of the experiment, and photographs were taken, always procuring to capture the full extension plane of each photographed fragment. A camera (Canon EOS 650 D, Canon Inc., Ota City, Tokyo, Japan) with an unchanged camera setting and the objective (EF-S 18-55 IS II Objective, Canon Inc., Ota City, Tokyo, Japan) were used to keep identical conditions such as the height of the lens above the floor (85 cm), height of tank above the floor (75 cm) and distance tank-camera (31 cm). The same photographic setup, with a constant distance to the specimens and a known size measurement reference were used every time the fragments were photographed. Subsequently, photographs were edited and processed using ImageJ software (version 1.44). Growth was derived by assuming the fragments to have the shape of a cylinder. Both length and the width (diameter) of the fragment were measured. As the fragment's diameter was slightly different throughout its length, the top, middle and bottom of the fragment were measured to calculate an average. Finally, changes in the coral fragments' surface area were calculated over time.

## Data analysis

Data analyses were carried out using the computing software R version 3.5.2 (*R Core Team, 2018*) and Rstudio version 1.1.456 (*RStudio Team, 2016*) and the R package "Lme4" from *Bates et al. (2015)*. In order to check whether there were any significant differences among the treatments, a Linear Mixed-Effects Model (LMM) was used for $O_2$ fluxes and growth, whilst a simple Linear Model (LM) was used to evaluate the Symbiodiniaceae cell densities. After an outlier treatment was performed, LMM models that suited the data were calculated and verified using model diagnostics: *i.e.*, model fit quality was carefully assessed using Pearson's residuals variance plots for each parameter together with linearity checks of the factors tested during the model's construction. The best model was chosen according to results obtained from direct model comparisons *via* ANOVA type II, alongside AIC criterion for additional comparison and best model selection. To estimate the significance of the fixed factors, we implemented an ANOVA type III for $O_2$ fluxes and Symbiodiniaceae cell densities, and an ANOVA type II for growth (*Zuur et al., 2009*). Corresponding approaches for degrees of freedom approximation were used accordingly (*Kuznetsova, Brockhoff & Christensen, 2014*). When *p*-values were determined to be $p < 0.05$, fixed factors were considered statistically significant. Whenever significant

differences were found, a corresponding *post hoc* Tukey test was executed using the R package "emmeans" by *Lenth et al. (2019)*. In further detail, the analyses were performed independently for each stage, ensuring consistency with the experimental design, *i.e.*, we created a LMM dedicated to the first stage: individual DOC addition in 12 tanks, with DOC (four levels), time and the interaction thereof as fixed factors. Additionally, a second model was created for the second stage: with DOC addition and increased temperature in 16 experimental tanks, DOC (five levels), temperature, and the interaction of DOC and temperature as fixed factors. The aquaria tank's information (*i.e.*, the corresponding tank identity) was included as a random factor in all our LMM models to account for additional sources of noise or unwanted variation related to differences among tanks. Furthermore, the donor colony identity information was included as random factor in the growth assessment models but not in the O$_2$ fluxes or the Symbiodiniaceae cell density models, always favoring model fit and statistical power. Hence, caution is advised when interpreting these results, as the models excluding this factor do not consider colony slope variations. As the four temperature control tanks were solely utilized for the second stage of the experiment, they were only included in the statistical analysis thereof. In addition, except for flow rate related parameters, the measured system characteristics of these four tanks did not differ statistically when compared to those of the DOC control tanks nor to the maintenance tanks (see Material S2).

## RESULTS

### DOC enrichment

For *P. flava* fragments exposed to different DOC concentrations, neither DOC, time nor the interaction between DOC and time showed any significant effect on the O$_2$ fluxes (LMM; $p > 0.05$; Table 1). The O$_2$ production rates showed mean values that varied from a minimum of 1.06 ± 0.37 mmol O$_2$ cm$^{-2}$ h$^{-1}$ to a maximum of 1.32 ± 0.41 mmol O$_2$ cm$^{-2}$ h$^{-1}$ by the end of the first stage of the experiment (Fig. 1A). The O$_2$ consumption rates showed a stable trend over time where mean values recorded at the end of the first stage were within a minimum of 0.65 ± 1.17 mmol O$_2$ cm$^{-2}$ h$^{-1}$ and a maximum of 1.61 ± 1.22 mmol O$_2$ cm$^{-2}$ h$^{-1}$ (Fig. 1B). Cell density was measured only at the end of the experiment when all treatments had already experienced the same temperature increase. Furthermore, growth showed no significant differences when exposed to individual DOC effects (LMM; $p = 0.19$; Table 2).

### Increased temperature

When *P. flava* fragments were exposed to simulated warming scenarios, O$_2$ production rates showed a decreasing trend towards higher temperatures (Fig. 2A). In contrast, O$_2$ consumption rates were not significantly affected by the increased temperatures (Fig. 2B; $p = 0.81$; Table 3). In particular, O$_2$ production was strongly reduced under increased temperature when compared to the temperature control. The factor temperature had a significant effect on O$_2$ production (LMM; F = 6.95, $p = 0.001$; Table 3), and subsequent pairwise temperature comparisons were significant only for the contrast between 26 and 32 and 28 and 32 °C ($p = 0.0024$ and $p = 0.0021$) respectively (Material S3.1). Moreover, the

Table 1 Linear mixed-effects model for $O_2$ production and consumption rates (mg $O_2$ m$^{-2}$ h$^{-1}$) of *P. flava* corals under individual DOC addition. Type III analysis of variance with Satterthwaite's approximation method for degrees of freedom.

| Factor | $O_2$ consumption | | | $O_2$ production | | |
|---|---|---|---|---|---|---|
| **Fixed effects** | *df* | F | *p* | *df* | F | *p* |
| DOC | 3 | 0.6482 | 0.6058 | 3 | 0.9971 | 0.4422 |
| Time | 4 | 0.6109 | 0.6578 | 4 | 1.3105 | 0.2871 |
| DOC × Time | 12 | 0.6796 | 0.7579 | 12 | 1.5900 | 0.1445 |

temperature factor alone did not have a significant effect on *P. flava* growth ($p = 0.56$; Table 2). However, a significant reduction of Symbiodiniaceae cell density was observed for the fragments exposed to heat stress at 32 °C at the end of the experiment (LM; $p = 5.35e−10$; Fig. 3).

### DOC enrichment and warming

For the second stage of the experiment, *i.e.*, where the gorgonian fragments were exposed to DOC enrichment and warming, neither the DOC treatment nor the interactions between DOC and temperature had a significant effect on the *P.flava* fragments' $O_2$ production or consumption (Table 3). However, there was a significant effect in the interaction between DOC and temperature on growth (LMM; $p = 6.1e−06$; Table 2; Fig. 4). The *post hoc* test with fixed DOC factor intercept varying across temperatures highlighted significant differences across the low (10 mg L$^{-1}$) DOC treatment ($p = 0.0046$) and the temperature control condition ($p = 0.0091$) at 26 °C (see Material S3.2). In detail, the significantly different contrasts included some of the pair combinations between the temperature control condition and the DOC control at 30 °C and, the temperature control condition and the low (10 mg L$^{-1}$) DOC treatment at 32 °C (see Fig. 4 and Materials S3.3, S3.4 and S4 for further details).

## DISCUSSION

### Effects of DOC concentration enrichment

Our results showed that individual DOC enrichment did not alter $O_2$ fluxes, or growth in *P. flava* fragments at any DOC concentration. Our findings contrast from previous scientific evidence conducted on hard coral species such as *Orbicella annularis* and *Pocillopora verrucosa*, which showed negative responses toward DOC enrichment since it likely triggers microbial activity, a consequent stoichiometric shift in the N:P ratio, and potential bleaching response. For instance, DOC concentration enrichment can initiate bleaching responses in hard corals (*Kuntz et al., 2005*; *Kline et al., 2006*; *Haas et al., 2016*; *Pogoreutz et al., 2017*; *Morris et al., 2019*). Nevertheless, our results match with recent studies conducted on the pulsating octocoral *X. umbellata* which showed no negative responses to individual DOC treatments regardless of their concentrations at the ecophysiological level (*Simancas-Giraldo et al., 2021*). Soft corals may display higher

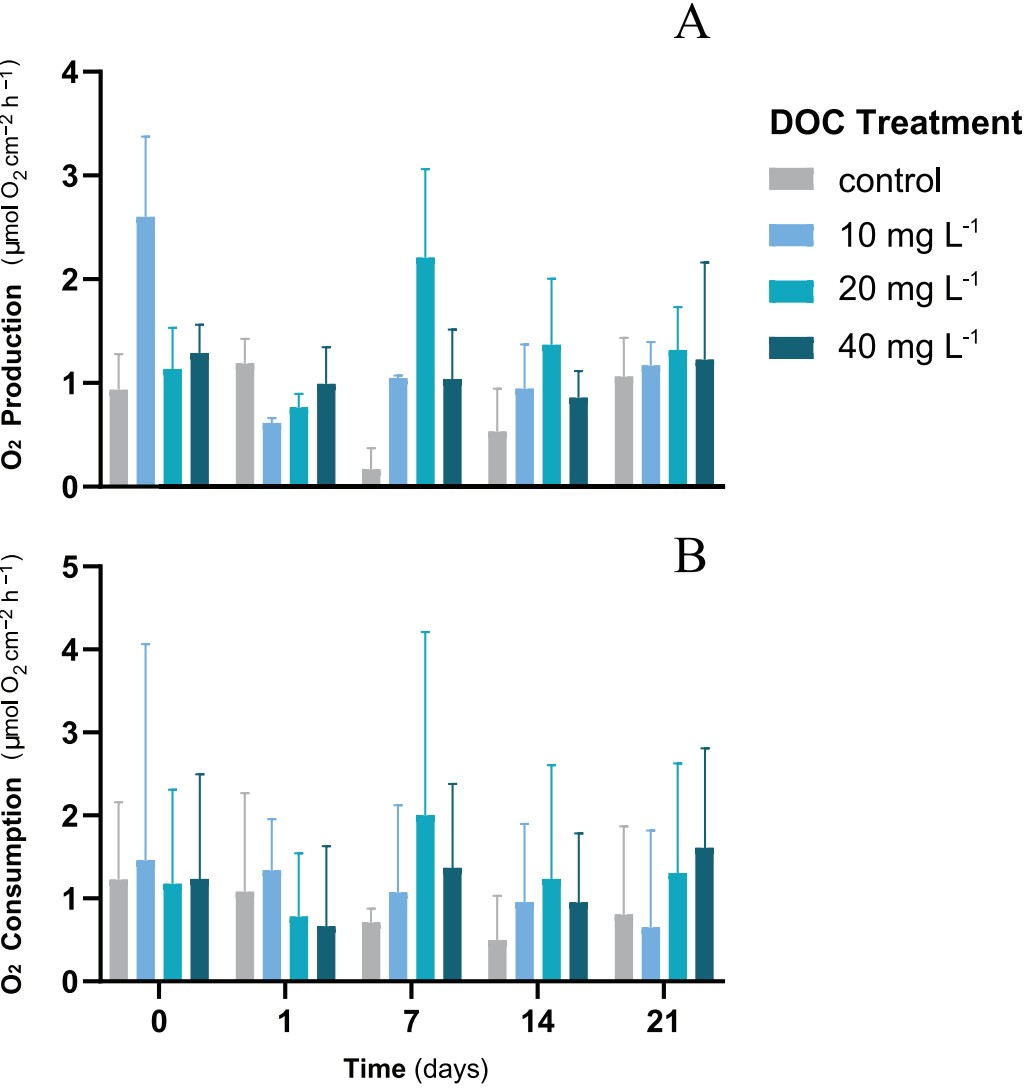

**Figure 1** **P. flava O₂ fluxes response to individual DOC addition.** O₂ production (A) and consumption (B) rates (mg O₂ m⁻² h⁻¹) of *P. flava* under simulated DOC organic eutrophication over time. The control: 2–3 mg L⁻¹ (grey), and the treatment conditions low: 10 mg L⁻¹ (light blue), medium: 20 mg (teal) and high: 10 mg L⁻¹ (dark blue) are represented accordingly. Individual DOC enrichment did not alter O₂ fluxes of the *P. flava* fragments at any of the DOC treatment concentrations assessed. Bars values indicate mean ± s.e.m. for *n* = 3 corals per treatment.               

tolerance to enriched DOC concentration in the water than hard corals, by either up taking the available DOC as an additional source of energy *via* the host (*Fabricius & Klumpp, 1995*), or by regulating internal nutrient availability of the holobiont *via* the role that the host-associated bacterial communities (*e.g.*, denitrifying bacteria) might play under changing environments (*Xiang et al., 2021*).

## Effect of temperature

Regarding temperature increases, the findings of this study on *P. flava* resemble the results observed in our previous works conducted on *X. umbellata*. The latter coral species is not

**Table 2 Linear mixed-effects model for *P. flava* corals growth as change in surface area. Type II analysis of variance with Satterthwaite's approximation method for degrees of freedom.**

| | | Growth | | |
|---|---|---|---|---|
| Fixed effects | Experimental stage | df | χ2sq | p |
| DOC | 1 | 3 | 4.490 | 0.1967 |
| DOC | 2 | 4 | 8.758 | 0.05701 |
| Temperature | 2 | 3 | 2.431 | 0.564 |
| DOC × Temperature | 2 | 12 | 48.949 | **6.1e−06***** |

Note:
Significant results where $p < 0.05$ are highlighted here in bold letters and indicated with asterisks (*). The asterisks indicate the $p$-value significance code accordingly.

sensitive to DOC enrichment, but shows a negative response to warming, though milder when compared to the present study (*Vollstedt et al., 2020*; *Simancas-Giraldo et al., 2021*; *Xiang et al., 2021*). Specifically, we found no effect of warming on *P. flava's* $O_2$ consumption rates, regardless of the temperatures reached during the warming stage of our study. However, we observed significant differences in $O_2$ production rates when subjected to warming. In particular, the rising water temperatures during the second stage of the experiment negatively affected $O_2$ production rates in *P. flava* at 32 °C. Such results contrast with studies performed on other gorgonian species from the Caribbean region such as *e.g.*, *Eunicea fleaxuosa* or *Eunicea tourneforti* (*Goulet et al., 2017*), but align with the well-known negative trends observed on several hard coral species in response to thermal stress (*Monroe et al., 2018*; *Ziegler et al., 2019*). Both decreased $O_2$ production and maximum quantum yield are among the first reactions of hard corals to thermal stress. This may lead to the dysfunction of photoprotective mechanisms and impair $CO_2$ fixation of the coral-associated Symbiodiniaceae; thus, causing bleaching responses (*Jones et al., 1998*). In general, thermal stress is a widely reported factor to cause bleaching in both hard corals (*Cardini et al., 2016*; *Hughes, Kerry & Simpson, 2018*; *Ziegler et al., 2019*) and octocorals (*Strychar et al., 2005*; *Slattery, Pankey & Lesser, 2019*), including gorgonians (*Lasker, 2003*; *Harvell et al., 2001*; *Prada, Weil & Yoshioka, 2010*; *Rossi et al., 2018*). Moreover, the observed reduction of $O_2$ production under high-temperature conditions aligned with the decreased Symbiodiniaceae cell densities observed at 32 °C may have been signaling the onset of an early bleaching response in our *P. flava* coral colonies. The loss of Symbiodiniaceae cells due to environmental stressors such as elevated seawater temperatures, may lead to symbiosis breakdown which in turn causes coral bleaching (*Brown, 1997*; *Fitt et al., 2001*; *Lesser, 2011*; *Karim, Nakaema & Hidaka, 2015*). However, despite the significant drop in Symbiodiniaceae cell density during the warming phase, the fragments appeared only moderately bleached, and no mortality was observed. This suggests a given potential for thermal tolerance until a certain threshold, that might be higher than that of many hard coral species reportedly sensitive towards warming (*Hughes, Kerry & Simpson, 2018*; *Ziegler et al., 2019*). Explanations for this relative tolerance could be related to many aspects of the coral holobiont including (but not limited to), species specific traits (*e.g.*, resistant bacterial or microbial communities), capability to modify holobiont parameters (*Goulet et al., 2017*; *Xiang et al., 2021*), differences in colony

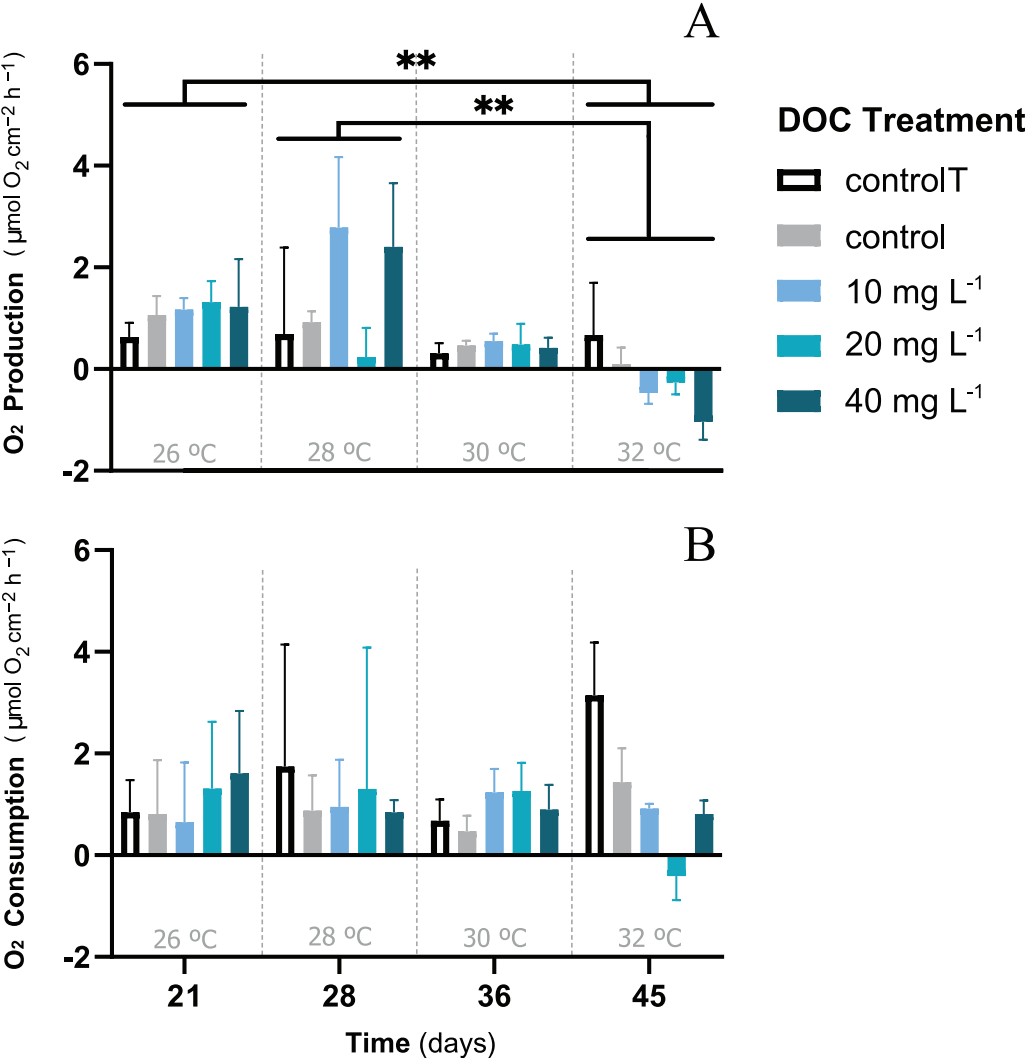

**Figure 2 *P. flava* O$_2$ fluxes response to DOC enrichment and warming.** *P. flava* response in terms of (A) O$_2$ production and (B) consumption rates (mg O$_2$ m$^{-2}$ h$^{-1}$) to increased temperature and prolonged DOC addition over time. The temperature control: 2–3 mg L$^{-1}$ at 26 °C (white), the combined increased temperature treatments including the DOC control: 2–3 mg L$^{-1}$ (grey), and the DOC treatments, low: 10 mg L$^{-1}$ (light blue), medium: 20 mg (teal) and high: 10 mg L$^{-1}$ (dark blue) are represented accordingly. Under increased temperatures O$_2$ production rates were negatively affected at 32 °C, and significantly different for the contrasts between 26–32 °C and 28–32 °C. However, no significant effects were found on *P. flava* O$_2$ consumption rates, regardless of temperature. Asterisks mark statistically significant differences ($P < 0.05$; LMM), and the quantity of asterisks displayed indicate the corresponding *p*-value significance codes. The bars values indicate mean ± s.e.m. for *n* = 3 corals per treatment.

morphology (*Conti-Jerpe, Pawlik & Finelli, 2022*) or potential adjustments of nutritional behaviors (*Grottoli, Rodrigues & Palardy, 2006*). Further, as highlighted by *Wooldridge (2014)*, many aspects of coral bleaching cannot be explained solely by the loss or persistence of algal symbionts amongst coral species but also by other host coral traits (*e.g.*, metabolic rates, heterotrophic feedings capacity) which are also believed to influence the thermal tolerance. Thus, all these aspects should deserve further exploration in future

**Table 3** Linear mixed-effects model for $O_2$ production and consumption rates (mg $O_2$ m$^{-2}$ h$^{-1}$) of *P. flava* corals under DOC enrichment and warming. Type II analysis of variance with Satterthwaite's approximation method for degrees of freedom.

| Factor | $O_2$ consumption | | | $O_2$ production | | |
|---|---|---|---|---|---|---|
| **Fixed effects** | *df* | F | *p* | *df* | F | *p* |
| DOC | 4 | 0.7785 | 0.56 | 4 | 0.8370 | 0.5310 |
| Temperature | 3 | 0.3254 | 0.8069 | 3 | 6.9528 | **0.0010**\*\* |
| DOC × Temperature | 12 | 1.9910 | 0.0592 | 12 | 1.2085 | 0.3199 |

Note:
Significant results where $p < 0.05$ are highlighted here in bold letters and indicated with asterisks (\*). The asterisks indicate the *p*-value significance code accordingly.

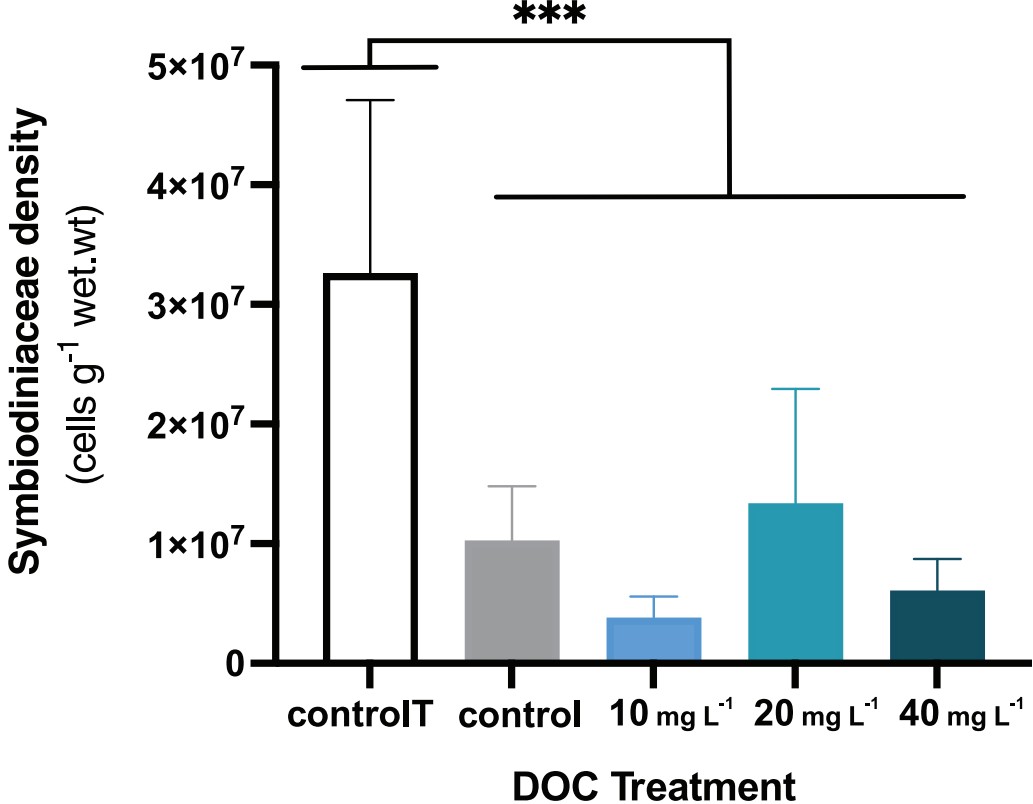

**Figure 3** *P. flava* **Symbiodiniaceae density in response to DOC enrichment and warming.** *P. flava* Symbiodiniaceae cell densities (cells g$^{-1}$ wet.wt) by the end of the experiment (day 45), corresponding to increased temperature of 32 °C and prolonged DOC addition. The graph presents the temperature control: 2–3 mg L$^{-1}$ at 26 °C (white), the combined increased temperature treatments including the DOC control: 2–3 mg L$^{-1}$ (grey), and the DOC treatments, low: 10 mg L$^{-1}$ (light blue), medium: 20 mg (teal) and high: 10 mg L$^{-1}$ (dark blue). Significant reduction of Symbiodiniaceae cell densities was observed by the end of the experimental term when increased temperature reached 32 °C, for all the treatments where the fragments had been exposed to increased temperatures. Asterisks mark statistically significant differences ($P < 0.05$; LMM), and the quantity of asterisks displayed is proportional to the corresponding *p*-value significance code. The bars values indicate mean ± s.e.m. for n = 3 corals per treatment.

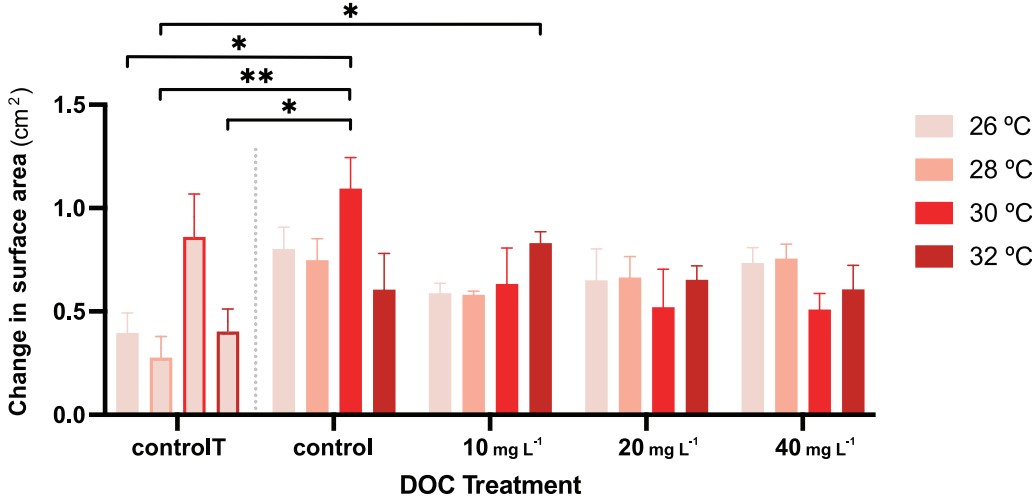

**Figure 4 P. flava growth as change in surface area in response to DOC enrichment and warming.** Coral surface area changes (cm²) corresponding to every temperature condition at each DOC treatment. The bar graph shows growth under individual DOC addition and warming as a coupled stressor (increasing red intensity scale). Temperatures per each DOC treatment are shown as 26 °C (light pink), 28 °C (salmon), 29 °C (red) and 32 °C (dark red). The first four bars correspond to the temperature control condition (controlT) with no DOC addition, constantly at 26 °C, but at each temporal step in which the rest of the system reached the corresponding temperature treatment targets. Thus, these four bars' outlines display increasing red intensities accordingly. While neither DOC nor temperature had any effect on growth, their interaction had a significant effect with the effect of DOC on growth varying depending on the temperature value. Asterisks mark statistically significant differences ($P < 0.05$; LMM) while the bars values indicate mean ± s.e.m. for $n = 3$ corals per treatment. The number of asterisks displayed on top of the lines indicate the corresponding $p$-value significance code.

studies in octocorals, including data on Symbiodiniaceae identity which likely would be needed to improve conclusions on the physiological responses of the octocoral holobiont.

Despite the experiments by *Wooldridge (2014)* being conducted on hard coral species, the occurrence of certain host traits in octocoral species may justify their potential resistance to thermal stress. In addition, despite the contribution of the Symbiodiniaceae to the energy budget of octocorals being species-specific (*Sorokin, 1991*), overall, it may be lower in some octocoral species compared to hard corals (*Baker et al., 2015*; *Ferrier-Pagès et al., 2015*). Furthermore, environmental parameters such as *e.g.*, water flow regimes or flow speed have also been shown to affect bleaching resilience, growth, mortality and to enhance coral feeding (*Nakamura, Yamasaki & Van Woesik, 2003*). In this study, we did not measure water flow speeds directly and water flow regimes significantly varied when comparing the maintenance tank to the rest of the experimental tanks in our system. Despite we controlled additional variation for this factor through our experimental manipulations and during our statistical analyses, flow rates discrepancy may have impacted our observed results, especially those concerning growth rates during the second stage of our experiments. Thus, the interpretation of our findings should consider these additional potential effects on coral tolerance related responses. Moreover, some octocoral species display higher trophic plasticity concerning nutrition compared to hard coral

species (*Schubert, Brown & Rossi, 2017*). This has been related to a higher capacity to cope with stress conditions as Symbiodiniaceae loss may not necessarily lead to a significant change in the coral energy input, preventing some octocorals from starving and dying (*Goulet et al., 2017*; *Schubert, Brown & Rossi, 2017*).

On the other hand, some hard corals show notable resilience capacity after bleaching by switching from acquiring fixed carbon *via* primarily photoautotrophic means to primarily heterotrophic means (that is, feeding) (*Grottoli, Rodrigues & Palardy, 2006*), a response that may occur also among octocoral species (*Schubert, Brown & Rossi, 2017*; *Lasker et al., 2020*). Although we did not assess this through our study, we speculate that *P. flava* could potentially compensate the decreasing Symbiodiniaceae density and lower photosynthetic activity by either modifying or regulating additional holobiont-related parameters such as *e.g.*, bacterial community structure, photosynthetic pigment activity or heterotrophic capacity (*Goulet et al., 2017*; *Xiang et al., 2021*; *Schubert, Brown & Rossi, 2017*). This would allow *P. flava* to effectively cope with increasing water temperatures under a diminished Symbiodiniaceae community and adverse environmental conditions eventually gaining advantages over other coral species such as hard corals until a certain threshold is met.

Moreover, these hypotheses may also contribute to explain the evidences of community phase shifts from hard coral-dominated towards octocoral-dominated reefs, as reported by recent studies in the Indo-Pacific as well as Caribbean regions (*Hoegh-Guldberg et al., 2009*; *Enochs et al., 2015*; *Lasker et al., 2020*). However, as shown in this study, the elevated temperature still implies negative consequences for the ecophysiology of *P. flava* (*i.e.*, decreased photosynthetic activity, moderate bleaching response) which will likely aggravate if conditions persist in the future, as seen in previous works on octocorals (*Lasker, 2003*; *Harvell et al., 2001*; *Prada, Weil & Yoshioka, 2010*; *Rossi et al., 2018*).

## Effects of DOC concentration enrichment and warming

The present study is the first to investigate the individual and combined ecophysiological effects of DOC enrichment and warming on the tropical gorgonian species *P. flava*. Our findings show that the interaction between DOC concentrations and temperature did not affect *P. flava* $O_2$ fluxes, while a significant effect was observed for growth. Growth was subsequently shown to respond differentially to determined DOC concentrations depending on the temperature, a feature that may confer an advantage or disadvantage to the coral under stress scenarios depending on the DOC concentration-temperature combination experienced. Furthermore, DOC concentration enrichment did not increase sensitivity to warmer temperatures in *P. flava* which contrasts with our hypothesis but aligns with our previous findings for *X. umbellata* corals (*Vollstedt et al., 2020*; *Simancas-Giraldo et al., 2021*). When compared to hard corals, these results suggest that *P. flava* may behave differentially towards the individual effects of DOC enrichment and warming, especially when bleaching response and growth are considered. Moreover, there was no overall effect of either temperature or the DOC concentration factor on growth. However, there was a significant effect in their interaction, in which there was an increase at 32 °C under low (10 mgl$^{-1}$) DOC concentrations and the effect of DOC concentrations on growth varied depending on the temperature. It is likely that *P. flava* responds to these

factors through mechanisms that actuate *via* diverse photochemical adjustments or physiological pathways, which should be further investigated in the future.

## CONCLUSIONS

Our findings suggest that the gorgonian species *P. flava* is not affected by individual DOC enrichment, in contrast to many other hard coral species. We also observed a significant decrease in *P. flava* $O_2$ fluxes and Symbiodiniaceae cell densities under higher temperatures, together with significant decreases in growth when subjected to elevated DOC concentrations and warming simultaneously. Thus, we advocate that the gorgonian octocoral in our study can still be negatively affected by increased water temperatures, despite showing substantial resistance to individual DOC enrichment. Nevertheless, DOC effects can vary depending on the temperature as observed in *P. flava's* growth.

The negative effects of expected climate change scenarios on this and potentially other octocoral species may lead to further structural simplification in coral reefs communities and ecological shifts towards alternative benthic assemblages which might gain competitive advantages (*e.g.*, macroalgae) under altered environmental conditions (*McManus & Polsenberg, 2004*; *deYoung et al., 2008*; *Sguotti & Cormon, 2018*; *Adam et al., 2021*). For these reasons, we suggest future studies to further explore the effects of combined local and global stressors on octocoral species *e.g.*, gorgonians, accounting for consequences of potential ecological transformation of coral reef communities' structure and productivity.

## ACKNOWLEDGEMENTS

The authors would like to thank Svea Vollstedt, Meghan Moger Kennedy, and Rassil Nafeh from the Marine Ecology Department at the University of Bremen, who provided technical assistance and support through our experiments.

### Funding

This work was supported by baseline funds from the University of Bremen, Department of Marine Ecology. There was no additional external funding received for this study. The funders had no role in study design, data collection and analysis, decision to publish, or preparation of the manuscript. The funders had no role in study design, data collection and analysis, decision to publish, or preparation of the manuscript.

### Grant Disclosures

The following grant information was disclosed by the authors:
University of Bremen.

### Competing Interests

The authors declare that they have no competing interests.

## Author Contributions

- Edoardo Zelli conceived and designed the experiments, performed the experiments, analyzed the data, prepared figures and/or tables, authored or reviewed drafts of the article, and approved the final draft.
- Susana Marcela Simancas-Giraldo conceived and designed the experiments, performed the experiments, analyzed the data, prepared figures and/or tables, authored or reviewed drafts of the article, and approved the final draft.
- Nan Xiang conceived and designed the experiments, performed the experiments, authored or reviewed drafts of the article, and approved the final draft.
- Claudia Dessì performed the experiments, analyzed the data, authored or reviewed drafts of the article, and approved the final draft.
- Nadim Daniel Katzer performed the experiments, analyzed the data, authored or reviewed drafts of the article, and approved the final draft.
- Arjen Tilstra analyzed the data, authored or reviewed drafts of the article, and approved the final draft.
- Christian Wild conceived and designed the experiments, authored or reviewed drafts of the article, and approved the final draft.

## Data Availability

The raw data are available in the Supplemental File.

## Supplemental Information

Supplemental information for this article can be found online at http://dx.doi.org/10.7717/peerj.14812#supplemental-information.

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
