# Peer review of "Individual and combined effect of organic eutrophication (DOC) and ocean warming on the ecophysiology of the Octocoral Pinnigorgia flava"

_PeerJ, doi:10.7717/peerj.14812_

## Round 0.1 · original submission · Major Revisions

Dear authors, I agree with the comments of the reviewers and would like to invite you to address each of the concern accordingly. In particular, a major issue was raised by Reviewer 2 regarding the possible division of data into three papers. The authors would need a strong justification to support the uniqueness of this specific data compared to the previous two. My other concerns have been beautifully pointed out by the reviewers. We look forward to your response.

Reviewer 1 ·

Basic reporting

Terms are not used consistently, such as "coral", "colony", "octocoral" etc, leading to confusion and unqualified statements. Some sentence fragments and grammatical errors exist.

References are generally OK if a bit old here and there.

Experimental design

Good from what I can see, but without donor colony information, Symbiodiniaceae data, and sampling/specimen data, paper is not replicable. Acceptance of this paper should hinge on these issues.

Validity of the findings

Good from what I can see, but without donor colony information and statistical treatment of this, Symbiodiniaceae identity data, and sampling/specimen data, paper is not replicable, and it is hard to judge validity of the authors' findings. Acceptance of this paper should hinge on these issues.

Additional comments

Overview: A nice experiment, this paper should be acceptable down the line. Still, as it is now, the paper has issues with the exact terminology used in the paper, leading to confusion, as well as missing data that should be provided to allow readers to better judge the work.

Major concerns:
1. Over-simplification – the authors have examined the effects of temperature and DOC on one gorgonian species. The experimental set-up is generally fine, the analyses seem appropriate, and the conclusions for this species are fine (if data for comments 2-4 are provided). However, the authors make many broad and sweeping statements on octocorals and scleractinian corals, often without qualifying statements, that I find to be over-generalizations. The authors have looked at ONE species of several hundred or thousands of octocoral species, and compared to scleractinians, which have ~800 species each of zooxanthellate and azooxanthellate species. The authors really need to be careful with their over-generalizations and make their phrasing more exact in order to not give readers false impressions. Please see, for example, lines 42-43; 302-303; 304-309; 314-315; 340-342; 345-348;
2. The authors used three donor-colonies, but I cannot see if they compared the effect of each colony, and this needs to be mentioned explicitly. If this was not done, then this reason also needs to be explained and explicitly discussed as to how this weakens your conclusions. See lines 145 and area.
3. Following the comment above, there is no information when and where and at what depth donor colonies were collected from, under what conditions, and no information on how species were identified, nor on their specimen numbers etc. All such information are needed to make this work replicable. See lines 145 and area.
4. No information is giving on the identity of the Symbiodiniaceae in this gorgonian species, nor on if identities were confirmed in donor colonies in this study. If there are no data on this, this remains a serious question that could very much affect the results. As it is now, one cannot judge much of the results due to this lack of data. Given all this physiological data, knowing symbiont identity is really important, as your data can inform on the physiology not only of the host octocoral but also symbiont(s) too.
Such M&M should go around line 176 or area I think.

Minor comments:
Line 55: At first mention, you should define what you mean by “coral”. Do you mean zooxanthellate scleractinians? Or anthozoans? In some areas your meaning of this word is different, so please be careful and define things clearly and be consistent in your usage of words. For example, line 276 means octocoral I think?
Lines 57-59: What are these effects? Negative? Positive? Explain in detail please.
Line 115: Where was seawater from? A kit? Or pumped in? Please add such information.
Line 154: I do not understand what an “accompany study” is or why it is relevant here.
Line 157: In this sentence, does “colony” mean “nubbin”? Please check, and be consistent in word usage throughout the paper.
Line 201 and other areas: Should “analysis” be plural? Check throughout the paper.
Line 258: This sentence should be past tense.
Lines 267-269: Sentence fragment. Also lines 313-314. The English really needs a careful check before resubmission.
Lines 289-291: I am confused. Gorgonians are octocorals, are you distinguishing them from soft corals? These are not really taxonomically valid terms, so please define your common names early on and stick with them?
References have some small mistakes, capitalization and other formatting.

·

Basic reporting

Basic Reporting is OK

Experimental design

Experiment design is also OK

Validity of the findings

no comment, see below

Additional comments

First, my apologies to the authors for the delay in getting the review done

After reading the manuscript and taking a look at the previous 2 manuscripts on Xenia umbellata by Simancas-Giraldo et al 2021 and Vollstedt et al 2020, I don't understand what was the necessity to divide the data into 3 papers including this one.
If I understood the methodology section in this manuscript, experiment on both the species was conducted at the same time? - if not then your explanation in the methods section is confusing

Line 152-154 - "Each experimental tank contained four frames, two holding n=10
153 P. flava and two holding n=10 X. umbellata fragments. The additional frames with X. umbellata
154 samples were used for accompanying studies (Simancas et al., 2021)"

If this is true, and as I can see from the results, the authors do not give any new information to the readers.

The results from this work show that the Pinnigorgia flava has a similar response to Xenia umbellata in terms of DOC and temperature stress. It would have been better to combine all the data from 2 species are write one comprehensive paper rather than unnessarily dividing into 3

If the result from previous papers and this work is not the same, then the authors have failed to explain it in this version of the manuscript.

So, unless the authors can explain the uniqueness of this work compared to the previous 2 papers, it is not ready to go to the next stage in my opinion

·

Basic reporting

Clarity of writing and English

I would suggest that the authors have this manuscript reviewed by a fluent English speaker to help improve the clarity. The language in the introduction is quite good and I was able to follow everything, but I started to have trouble understanding the results section. The discussion is a mixed bag, where some areas have language that is quite clear while others have some text that is difficult to understand. If the author are non-native English speakers, I want to commend them for their writing. I am impressed by their ability to write professionally in a second language, but I still think that the manuscript would benefit from input from a fluent speaker or professional review service. Some examples of text where language confused me are lines 78-179, 192-195, 236, and 301-302.

Additionally, I want to encourage the authors to use consistent language when discussing aspects of their experimental design. In particular, the way the authors refer to the different treatments and controls was very confusing. For example, on line 236 the authors say that net photosynthesis strongly decreased in the increased temperature control – the phrase increased temperature control made me think it was a control for increased temperature, but I believe the authors are referring to the DOC control that experienced increasing temperatures. Another example is on line 253 where the authors mention a difference between the low DOC treatment and the “none increased temperature control condition” – I think this is referring to the control that didn’t experience a temperature increase, but it’s not clear. I think the best way for the authors to address this is to clearly state or define the names for the different treatments and experimental aspects and consistently using them throughout the manuscript. Another example that I found confusing was the use of colony vs fragment, which sometimes seemed distinct but other instances seemed interchangeable.

Finally, the authors needs to switch from using "coral surface" to "coral surface area" or "change in surface area" where appropriate.


Literature references and background/context

The authors do a good job of presenting the core of the topics necessary to introduce their study. Specifically, I can see how they are justifying their work by explaining that gorgonians are ecosystem engineers that are playing a more prominent role on coral reefs as they become more abundant. Their response to multiple anthropogenic stressors is therefore important for understanding the future of reefs under global climate change. However, I think they need to expand the introduction further. For example, the introduction includes a paragraph summarizing the negative consequences of eutrophication on corals and coral reef communities, however the story is not this simple. There are many studies that show that some eutrophication can have positive impacts on corals (See Fabricius 2005 Effects of terrestrial runoff on the ecology of corals and coral reefs: review and synthesis for a nice review as a place to start). This is critical for the authors to include given that their results did not show an impact of DOC on gorgonian physiology and growth.

I would also like to see a more detailed introduction on the effects of bleaching and/or eutrophication specifically on octocorals and/or gorgonians given that this manuscript is focused on a member of this understudied group. Rather than just mentioning that octocorals or gorgonians have been looked at before, as on lines 58 and 85-86, it would be better to summarize previous findings, even if just to say that this group of corals responds similarly to more expansively studied scleractinian corals. Similarly, while this does indeed seem to be the first study examining the combined effects of warming and eutrophication on gorgonians, this has been investigated in other ocotocorals. These similar studies must be included in the intro.

The authors also chose to study DOC which they say is not as widely studied, but a summary of the affect of these compounds specifically (especially glucose) would be useful to readers. There is some nice text in the discussion explaining how DOC impacts coral, but it would be more appropriate to introduce this published work in the intro so that the discussion can focus more on a comparison between previous results and those of the current study.

Given the relatively broad readership of PeerJ, I think the authors should consider including some information about the difference between scleractinian, octocorals, and gorgonians as many non-experts are unfamiliar with these distinctions as well as some basic information about the symbiosis corals maintain with Symbiodiniaceae.

Finally, I would ask the authors to include previous work that has been conducted examining the combined effects of warming and eutrophication on octocorals in the introduction, including the data from X. umbellata from this experiment that is already published.


Article structure, figures, tables, and availability of raw data

As detailed above, there are sections of the discussion that introduce previously published work that should be brought up in the intro.

I believe an overhaul of the figures would greatly strengthen this manuscript. Including the following suggestions:

Given that the depict data collected through time, Figures 1 and 2 would be much more intuitive as point and line graphs as opposed to bar graphs. This will help the reader see how each treatment changed over the course of the experiment, which is currently difficult with bars.

Include significant results on the figures: indicate with stars or p values when there is a significant difference. This greatly helps the reader interpret the results.

Consider adding some indicator of temperature in the Figure 2 – perhaps some kind of arrow or scale underneath the x axis that reminds the reader how temperature was being increased over the duration of the experiment. When reading the results, the authors say that there is a significant difference between 26 and 32 deg C, yet on this figure it’s not clear which days those temperatures are on and the reader needs to go back and consult the methods to figure it out.

In figures 1 and 2, I suggest stacking the panels on top of each other as they share the same x-axis. I would also consider making the respiration rates negative to reflect oxygen consumption, or labeling the axis to indicate this (Oxygen production and oxygen consumption) to help the reader follow the figure more easily.

Rather than call your DOC treatments Low, Medium, and High, I would simply name them by concentration: 10, 20, and 40. This will make figure interpretation more straightforward for the reader.

I’m not certain, but I believe figure 4 is showing change in coral surface area (the initial surface area subtracted from the final surface area at the end of the experiment). This should be the label of the y-axis (change in surface area) and be explicit in the legend.

All raw data are provided in the supplemental materials, however there are several aspects of the data that are unclear including "Round" in the incubation dataset and "delta" in the surface area dataset. While not strictly necessary, it would also be helpful to have the two incubation data files and the two surface area files so that the results for each measurement are in one file together. It is unclear why they are currently each separated into two files.

The submission is appropriately self contained and the results presented are appropriate for the hypotheses as written, however in the line by line comments I provide below, I make some suggestions to improve the hypotheses. Additionally, there is one critical issue with the description of the methods that prevents me from evaluating whether the data presented assessing the affect of temperature are robust (also detailed below in the experimental design section).

Experimental design

This work is well withing the aims and scope of Peer J. I also believe the research question is well defined and fills a meaningful gap. One piece of information I would like added is how the authors set the DOC concentrations. These values seem quite high and outside what I would expect in field conditions, even at highly impacted sites. I think justification for this aspect of the design is important for the relevance of the study overall.

As far as I can tell, the experimental design is sound, although more information is needed both about the tank setup and the statistical analysis before this manuscript can be published. Specifically, there is not enough detail about the tanks used as controls for temperature. On line 126 the authors mention “Four new control tanks were provided as reference controls for temperature as soon as the temperature was increased”. Without further explanation, this is not clear enough for publication. How were the corals in these tanks treated prior to the point in the experiment? What was the replication in the tanks? What was the DOC concentration in these tanks? There is simply not enough information provided for me to assess the validity of this method. Finally, additional information is needed about the statistical analysis including the model diagnostics that were used to select and verify the model as well as the treatments tested with each model and how they were incorporated into the model.

Validity of the findings

Without further information about the control and statistical analysis (detailed above) I am unable to assess the validity of the temperature treatments. If the authors can clarify these aspects of their methods, then I suspect they will have a study suitable for publication, but this information must be provided before that can be determined.

Additional comments

I would ask the authors to avoid using the term “zooxanthellae” as this term is not specific to Symbiodiniaceae, but rather includes other dinoflagellates and diatoms. See the Symbiodiniaceae Style Guide by Parkinson et al for more information: https://www.thelifeaquatic.net/?page_id=292

I have also provided more detailed comments referencing line numbers with the intention of helping the authors improve the clarity of their manuscript. Some of this is repetitive with points already brought up, but I hope it is helpful to the authors to have specific sections of the text referenced:

Lines 35-38: Consider combining the two sentences on lines 35-38 so the are separated with a comma rather than a period

Line 46: suggest deleting “the” so it reads “In marine habitats”

Line 52: change from “it is demonstrated” to “it has been demonstrated”

Line 56: “damages” should read “damage”

Lines 57-59: I like how the authors bring up previous work investigating the impacts of warming on these two groups of corals, but it would assist the reader if they briefly compare/contrast the affects of warming on these two groups – do they respond the same? If not, can the authors summarize the differences?

Line 68: the second “via” is unnecessary, suggest deleting

Line 97: Is there any further justification for why P. flava is a model organism? Photosymbiotic gorgonians are less common in the Indo-Pacific, I’m curious if there is any other attributes that make is a good model.

Lines 98-108: I think it would be easier for the reader to follow if you introduce the overview of the experimental design first followed by the hypotheses. I also think you can consider reducing the experimental details you include here. All the reader needs at this stage is enough information about your experimental design to understand your hypotheses. I would also consider making your hypotheses more specific – what did you expect for each parameter you measured? For example, did you expect photosynthesis to decrease and respiration to increase under both stressors?

Line 114: I suggest changing the title of this subsection since you don’t discuss how the corals were prepared until the following section. I also would consider changing “Experimental Design” to something like “Experimental tank setup” since you introduce some aspects of the design later on (for example, colony replicates, fragment replicates, etc.

Line 115: Please spell out the genus here as this is the first time this species is mentioned in the manuscript.

Line 120: I suggest changing “temporal sections” to “time periods” – temporal sections was quite confusing to me

Line 125-126: You can delete this sentence, it doesn’t provide any new information or understanding to the reader.

Line 126-127: I am confused by this sentence – what do you mean when you say four new control tanks were provided? Did these have corals in them during the first time period? What were the DOC concentrations in these tanks – one tank per treatment, including the controls? Does this mean the total number of tanks in this portion of the experiment was 16?

Line 127-129: It sounds like these tanks were recirculating and not flow-through yet there is an outflow? This needs clarification.

Lines 127-142: I suggest moving the description of the tank system earlier, before you describe the two time periods of the experiment – this seems more clear and straightforward for the reader

Lines 135-137: You refer to a previous publication which presents the chemical parameters measured during the experiment – I think here is where you need to state that there were fragments of X. umbellata in each tank along with your P. flava. I was confused until I read more later on.

Line 145: suggest changing “size” to “length” (if appropriate) and deleted “from”

Line 152: The math here indicates that a total of 240 P. flava fragments were used in the experimental tanks (12 tanks x 20 frags per tank). This leaves 40 unaccounted for fragments (line 145), which I suspect were used in the “four new control tanks”, referenced on line 126 (so 10 per tank in these tanks?). If this is the case, I would consider bringing those up here. If the remaining 40 nubbins were not used, I would also mention that (or include the total number used in your experiment earlier). At this point in the manuscript I’m confused about the four new control tanks, how there were treated, and what went in them – I would try to write out explicitly what they were and any way they differed from the experimental tanks.

Line 157: I am a bit confused when the authors say “each coral colony (n = 3 per treatment) was transferred to individual incubation glass chambers”. It sounds like measurements were made when the colony was whole, prior to fragmentation. Do the authors mean instead that one fragment representing each colony from each treatment or tank was used for these measurements? It is unclear how many fragments were used in these measurements.

Line 163: Again, it sounds here like Pn and R measurements were each taken twice per day, but this seems quite high – perhaps the two times per day here indicates one for Pn and one for R? I would revise this for clarity.

Line 173-174: This should read “normalized to coral surface area” and the authors should detail how coral surface area was measured (perhaps via the growth rate measurements below?).

Lines 177-178: When the authors write that they “randomly selected P. flava colonies from each aquariua treatment (n = 3 per treatment) it is unclear to me if this means one fragment from each tank (as there were 3 tanks per DOC treatment) or three from each tank? Or three randomly chosen from the three tanks in each treatment pooled? Suggest clarification here.

Line 179: “weighted” should be “weighed”

Lines 180-181: Please provide details about how the host tissue was separate from the axis and homogenized – what tools were used?

Lines 181-183: Here, the authors say the resulting homogenates were quantified. I believe they mean to say the density of Symbiodiniaceae cells was quantified by hemocytometer counts of coral tissue homogenate – please revise.

Line 188: I believe Image should not be capitalized here

Lines 198-199: I suggest adding more detail here – what did you measure from your photographs? The length of the fragment, and then the width? Was the width measurement used as the diameter of your cylinder?

Line 199: Please change “surface changes” to “changes in sample surface area” or, even better, “changes in fragment surface area” for clarity.

Line 202: “R” seems to be missing from the reference in this line

Line 205: Suggest deleting “Analysis”

Lines 207-209: Please be more specific about how you chose and verified your model – what model diagnostics did you use?

Line 250: Here, the authors report a significant difference in the fragments surfaces – is this the change in surface area over the course of the experiment? The phrasing is unclear.

Line 312: this is the fist time you mention starvation as a consequence of bleaching. You introduce a lot of technical aspects of bleaching in other portions of the introduction but I would consider adding this as well. Since you have a whole paragraph on this, it may also be worth citing some papers that have looked at increased bleaching resistance and resilience in scleractinian corals (supporting your ideas here!) – Wooldridge (2014; Differential thermal bleaching susceptibilities amongst coral taxa: re-posing the role of the host) and Grottoli et al (2006; Heterotrophic plasticity and resilience in bleached corals) have nice papers on this.

Lines 331-333: Here I think it would be helpful to the reader to restate some of what has been found with hard corals – remind them what the literature has found so that the reader can understand how your results are different. It may also be worth mentioning studies where hard corals bleached under the similar conditions to your study – it is difficult to make comparisons compelling without indicating that the conditions were the same or similar.

Line 339: Be careful here – you only studied one species, so rather than saying your findings suggest octocorals are more tolerant to stressors, I would say your results contribute to the growing body of literature that supports this. You can’t make generalizations about octocorals from the one species you studied, but in the context of the literature it is compelling.

---

## Round 0.2 · Major Revisions

I agree with the reviewers that additional explanations of the experimental design are needed to ensure reproducibility. Do kindly address each comment/suggestion. Proofreading of the entire manuscript is also suggested to improve clarity.

Reviewer 1 ·

Basic reporting

Generally much better, the authors have revised the paper well.

Experimental design

Generally much better, the authors have revised the paper well.

Based on my previous comments on Symbiodiniaceae, I think adding a caveat to the Discussion that such data would be needed to better inform conclusions should be added.

Validity of the findings

Generally much better, the authors have revised the paper well.

Additional comments

Some comments that need addressing.
1. The introduction now introduces the octocorals and hexacorals, which is welcome, but this section includes some erroneous information. Gorgonians are a functional morphological grouping, but not a valid taxonomic grouping. Some gorgonians have axes of different substances. Thus, more accurate explanation and introduction is needed here. Words like "Anthozoans" should not be capitalized.
2. Even if these are cultivated octocoral colonies, where did they originate from? Source of colonies is needed.
3. "Cladocopium" needs to be italicized. Still, your sentence on line 249 does not really make sense to me. You did NOT identify the symbionts in your colonies, and thus you really need to explicitly say this, and say you assume they are Cladocopium sp. based on Goulet et al. 2008 or something like this.

·

Basic reporting

no comment

Experimental design

the authors have revised this part

Validity of the findings

the authors have revised this part

Additional comments

I don't have any more comments or suggestions

·

Basic reporting

The authors have made effort to improve the English and general language throughout the manuscript, however there are still three issues that need to be addressed:

1) English: In some areas, incorrect English still limits or impedes understanding of the content. Examples of locations where this occurred are lines 236-239, 330-333, 404-405, 423-425, and all figure and table legends.

2) Use of consistent nomenclature: the authors have improved the understandability of their methods and amended the names of different treatments, however there are still inconsistencies in how they refer to their different treatments. Throughout the manuscript, all of the following phrases were used to discuss different treatments: single DOC treatments, single DOC addition, individual DOC treatment, continuous DOC addition, DOC controls, control condition for temperature treatment, heat-stressed control, temperature control. Some of these seem to refer to the same treatment. The authors also use "high" and "low" to refer to different DOC concentration treatments in the results and discussion when these are introduced as "10 mg per L" and "40 mg per L" in the methods. The authors present a study with a somewhat complex design. It impedes the reader's ability to understand the manuscript when inconsistent names are used for treatments. Please use consistent names for treatments throughout all aspects of the manuscript.

3) Technical language: the authors still use the outdated term "zooxanthellae" in two locations in the manuscript. The authors also still use "surface" instead of "surface area" in some of the figure/table legends and supplemental materials.

The authors have used the previously provided comments to improve the introduction. Other than several more minor suggestions in the line by line comments, I think the background provided is sufficient.

The structure of the manuscript as well as the figures are improved, but the figure legends are incomplete and missing many previously suggested changes. Figures and figure legends are the most important part of an article - please write comprehensive legends so that the reader can understand what the figure shows without referring to the text. All data is provided, however in the data files there seem to be more than three colonies noted (1A, 3B, 7F, 8H). I suggest the authors include some text in the supplemental materials allowing the reader to properly interpret this column. In the incubation and surface area files, the temperature is incorrect for the controlT fragments (this increases the same as all the other treatments when I believe it should remain at 26). Finally, I suggest the authors change the treatments in the data files from low, medium, and high to 10, 20, and 40 to maintain consistency.

Despite there being two other publications from this experiment, I think that the data included in this manuscript represents enough for a self-contained unit of publication and all the results necessary to test the hypotheses are provided here.

Experimental design

As stated in my previous review, this manuscript is well within the aims and scope of PeerJ. The authors used previous comments to improve the question and hypotheses.

The authors have also greatly improved the methods section, which I think is much clearer. However, there are still a few areas where I am not sure if I follow the authors' meaning and could repeat the methods (specified in the line-by-line comments attached). Further, the authors have still not provided sufficient information about the large holding tank the temperature control fragments were kept in for me to feel confident that this is a valid control.

On lines 213-216 the authors state that experimental fragments were "propagated from three clonal mother colonies that were kept in a 420 L aquarium in the facility in a different aquarium for more than one year prior to the start of our experiment" and that they kept "the same conditions of this aquarium for all the control tanks used for this experiment". This statement is not enough to validate keeping the temperature fragments in this tank during phase one of the experiment, but rather the authors need to provide data on the different parameters measured in the large aquarium to ensure conditions were the same as in the DOC tanks. Perhaps these data are already provided in one of the other two papers published from this experiment - great! Please add a general statement saying this, listing the parameters that were measured and add the relevant citation here. If these data were not presented in the other pubs, the authors need to include it (in the supplemental materials would be fine).

I am particularly concerned about two parameters that could easily differ in the large tank: light and flow speed. Were the same types and numbers of lights used over the big donor tank and experimental tanks? If so, are the depths of the tanks different? If you use the same light source over different depths, the light the corals are exposed to will be different, and light plays a role in thermal stress response and bleaching. Luckily you report the light intensity (presumably experienced by the fragments, not at the surface of the water) for your experimental tanks so reporting the same metric (and if it’s similar) for the big donor tank will address this. Flow rate is not reported for your experimental tanks. If the big tank had a different flow regime than your DOC tanks, the temperature control fragments may have had to spend energy adjusting to the new flow regime when they were moved. Flow rate has also been shown to affect bleaching resilience and mortality (see Nakamura et al 2003 Water flow facilitates recovery from bleaching in the coral Stylophora pistillata; Grottoli et al 2020 Increasing comparability among coral bleaching experiments – this is a nice summary of what parameters matter and should be reported in bleaching/thermal stress studies). A large aquarium with other corals could host a plankton community that your fragments were able to eat more of if the flow was faster. Perhaps you can get some flow measurements of where the fragments were in the big tank and experimental tanks? Do you have any measurements of plankton communities in this tank? Is the water in this tank or your experimental tanks filtered? Through what size filter?

These may seem like details, but it is critical that you show, with data, that your control fragments experienced the same conditions as the experimental fragments during the first phase of the experiment (except for DOC concentrations of course). Any differences in conditions prior to phase two of the experiment could explain the differences you see in growth in the temperature control treatment.

Validity of the findings

The authors still need to provide data to justify the treatment of the temperature controls prior to phase two of the experiment before I can assess the validity of the findings. I would like to see (or be referenced to) data on the conditions in the large holding tank the temperature controls were kept in during phase one of the experiment.

The authors make some leaps in their conclusions. I do not think they have evidence to support increased heterotrophy in P. flava during the experiment (lines 406-407) nor heterotrophy preventing bleaching (lines 407-408).

Additional comments

Please see the attached PDF for line-by-line comments. Many of these are minor and not strictly necessary, but aim to improve clarity. I amended many, but not all instances where language was inhibiting understanding.

---

## Round 0.3 · Minor Revisions

I thank the authors for following through three rounds of review. Kindly address the minor concerns from the reviewer and I look forward to receiving the final manuscript. Do also incorporate a round of proofreading to ensure grammar consistency throughout the manuscript.

·

Basic reporting

The authors have clarified the vast majority of the manuscript. In the line-by-line comments, I point out a few remaining instances where language is muddling meaning. There are some areas with grammatical errors that I did not point out, but the meaning is clear.

The authors have revised their manuscript to include the outdated, no longer accepted order Gorgonacea (or gorgonacean; see title, line 50, and throughout the text). This order has not been used since 1981 when Bayer merged it with three others into Alcyonacea (https://www.biodiversitylibrary.org/part/45968). Please remove all uses of this order from the manuscript, including from the title. McFadden et al. 2010 (https://doi.org/10.1093/icb/icq056) is a great resource that explains much of the taxonomic history of octocorals. For the most up to date systematic structure of Octocorallia, please refer to the recent publication by McFadden et al. (2022; https://doi.org/10.18061/bssb.v1i3.8735).

I suspect that the authors are using Gorgonacea as a replacement for the word “gorgonians”. While gorgonians are not a recognized taxonomic group, it is useful for referring to soft corals that have a more extensive 3D structure as a result of their axial skeletons. I think it would be fine for the authors to use “gorgonians” in their manuscript, but they must explain what this word refers to for clarity, especially for readers unfamiliar with the complex classification of soft corals.

Experimental design

The authors should acknowledge in their methods that the flow rate was different in the parental colony tank where the temperature controls were held for the first phase of the experiment. Please see the line-by-line comments for details.

Validity of the findings

No comment

Additional comments

I commend the authors for applying constructive feedback on their manuscript and persisting through three rounds of review!

---

## Round 0.4 · Minor Revisions

I thank the authors for following through with the revision process. I think the authors have sufficiently addressed all of the concerns raised by reviewers. However, I think the manuscript would benefit from one last round of careful proofreading. I have also pointed out some obvious ones below:

Line 170: change 'summarizing' to 'In summary'
Line 249: 'frags'? as in 'fragments'? If yes, I would suggest using the full term to provide a clearer understanding.
Line 393: remove 'e.g.,', it is redundant.

---

## Round 0.5 · accepted · Accept

I believe the authors have addressed all concerns raised regarding this manuscript and it is now ready for publication.